

# Co-occurrence patterns of litter decomposing communities in mangroves indicate a robust community resistant to disturbances

Rodrigo G. Taketani[1,2], Marta A. Moitinho[2], Tim H. Mauchline[3] and Itamar S. Melo[2]

[1] Department of Soil Sciences, "Luiz de Queiroz" College of Agriculture, University of São Paulo, Piracicaba, SP, Brazil

[2] Laboratory of Environmental Microbiology, Embrapa Environment, Brazilian Agricultural Research Corporation-EMBRAPA, Jaguariuna, SP, Brazil

[3] Sustainable Agriculture Sciences, Rothamsted Research, Harpenden, United Kingdom

Corresponding author
Rodrigo G. Taketani,
rgtaketani@usp.br,
rgtaketani@gmail.com,
rgtaketani@yahoo.com.br

## ABSTRACT

**Background.** Mangroves are important coastal ecosystems known for high photosynthetic productivity and the ability to support marine food chains through supply of dissolved carbon or particular organic matter. Most of the carbon found in mangroves is produced by its vegetation and is decomposed in root associated sediment. This process involves a tight interaction between microbial populations, litter chemical composition, and environmental parameters. Here, we study the complex interactions found during litter decomposition in mangroves by applying network analysis to metagenomic data.

**Methods.** Leaves of three species of mangrove trees typically found in the southeast of Brazil (*Rhizophora mangle, Laguncularia racemosa,* and *Avicennia schaueriana*) were collected in separate litter bags and left on three different mangroves for 60 days. These leaves were subsequently used for metagenome sequencing using Ion Torrent technology. Sequences were annotated in MG-RAST and used for network construction using MENAp.

**Results.** The most common phyla were Proteobacteria (classes Gamma and Alphaproteobacteria) followed by Firmicutes (Clostridia and Bacilli). The most abundant protein clusters were associated with the metabolism of carbohydrates, amino acids, and proteins. Non-metric multidimensional scaling of the metagenomic data indicated that substrate (i.e., tree species) did not significantly select for a specific community. Both networks exhibited scale-free characteristics and small world structure due to the low mean shortest path length and high average clustering coefficient. These networks also had a low number of hub nodes most of which were module hubs.

**Discussion.** This study demonstrates that under different environmental pressures (i.e., plant species or mangrove location) the microbial community associated with the decaying material forms a robust and stable network.

## INTRODUCTION

Mangroves are highly productive coastal ecosystems (*Holguin, Vazquez & Bashan, 2001*) contributing 10–15% of the global coastal carbon storage (*Alongi, 2014*). Most of this organic matter (OM) is stored in their sediments (*Alongi, Boto & Tirendi, 1989*; *Siikamäki, Sanchirico & Jardine, 2012*). However, a significant part of that is exported to surrounding environments as dissolved or as particular OM (*Alongi, Boto & Tirendi, 1989*). Most of the carbon found in mangroves is produced by vegetation, although there is also a contribution of water column and sediment organisms to the carbon stock (*Holguin, Vazquez & Bashan, 2001*; *Kristensen et al., 2008*).

Litter degradation is a complex multifactorial process dependent on litter chemical composition, environmental parameters and populations of both macro and microorganisms (*Schneider et al., 2012*). This process is initiated and mediated by litter colonizing microbes (*Heijden et al., 2016*; *Purahong et al., 2016*).

Efficiency of litter degradation and decomposition is closely linked to organic matter lability (*Kristensen et al., 2008*; *García-Palacios et al., 2016*) with differences leading to the formation of separate niches occupied by specific microbes (*Frossard et al., 2013*). This dynamic process leads to complex interactions between populations with different metabolic capabilities and ecological functions (*Dini-Andreote et al., 2014*). These interactions can lead to the formation of patterns of co-occurrence (and co-exclusion) between populations that could unveil ecological processes yet unknown (*Green et al., 2017*). The use of network analysis has unveiled the relationships between populations and functions in the most diverse processes and habitats (*Faust & Raes, 2012*). The interactions between microorganisms happen in a variety of ways such as the flow of energy, matter, and signals leading to the formation of complex ecological networks (*Montoya, Pimm & Sole, 2006*). Studying these dynamics is essential to understand the processes that govern ecological networks (*Zhou et al., 2011*; *Faust & Raes, 2012*; *Deng et al., 2016*). In litter decomposition, the interactions between complementary microbes is required for the decomposition of complex polymers such as cellulose and lignin (*Purahong et al., 2016*).

This study was designed to study the complex interactions observed during litter decomposition in mangroves by applying network analysis to metagenomic data and to test the hypothesis that in such a complex environment the ecological network formed by these communities is responsible for the homeostasis of the process.

## MATERIALS AND METHODS

### Study site and field experiment

This study was conducted in three different mangrove sites in the State of São Paulo, Brazil. One in the south of the state located in the city of Cananéia (Can) (25°05′03″S–47°57′75″W) and two in the city of Bertiga (Bert and BC) (23°54′08″S–46°15′06″W and 23°43′74″S–47°57′75″W, respectively) in the center of the state. The former (Can) is a preserved mangrove with no history of anthropogenic impact; the other two are located close to highly urbanized and industrial areas and therefore have high human influence (*Andreote et al., 2012*). Also, BC had a major oil spill in 1983 from which is still in process

of recovery (*Andreote et al., 2012*). In each of these mangroves, fresh and mature leaves (at the same phenological state) from the three main species of mangrove trees (*Rhizophora mangle, Laguncularia racemosa,* and *Avicennia schaueriana*) were sampled directly from the tree. The chemical composition of these leaves varies between species, *R. mangle* has the lowest hemicellulose and protein content and the highest lignin content, *L. racemosa* has the highest hemicellulose, and the lowest cellulose content while *A. schaueriana* has the highest cellulose and protein content (see *Moitinho et al., 2018* for details). Field sampling was approved by the Biodiversity Authorization and Information System (SISBIO #20366-3).

Sampled leaves were added to sterile nylon litterbags (25 × 25 cm, mash size 0.1 mm) containing 300 g of each plant material and left over the sediment for 60 days during the fall of 2014 (10th of March to 8th of may). For each plant species, four different litterbags were randomly distributed in a 30.0 m² area in each mangrove forest. After this time, the bags were collected and 100 g of decomposed material was immediately frozen in liquid nitrogen for DNA extraction.

## Nucleic acid extraction, processing, and sequencing

The total DNA was extracted from the decaying leaves using the RNA PowerSoil® Total RNA Isolation Kit and the RNA PowerSoil® DNA Elution Accessory Kit, respectively, following the manufacturer's protocol. DNA quality and quantity were evaluated with the Nanodrop 2000 and by 1% agarose gel electrophoresis. Metagenomic libraries were constructed using the Ion Xpress Plus Fragment Library Kit with Ion Xpress Barcode Adapters following the manufacturer's protocol. Sequencing templates were constructed with Ion PGM Template OT2 400 Kit in an Ion Torrent OneTouch 2 equipment. Sequencing was performed using the Ion PGM 400pb Sequencing Kit on an Ion Torrent Personal Genome Machine. The sequencing of 21 libraries obtained a total of 9.317.861 reads with an average of 233 bp.

## Sequencing processing and annotation

Metagenomic sequences were uploaded to MG-RAST and were processed using the default parameters and can be found under project number mgp13300. All phylogenetic analysis presented here is the result of the Best Hit Classification against the M5NR database using an $E$-value cut-off of $10^{-5}$, a minimum identity of 60% and a minimum alignment of 50 bp (*Delmont et al., 2011*). The functional annotation was performed by Hierarchical Classification against the Subsystems database using an $E$-value cut-off of $10^{-5}$, a minimum identity of 60% and a minimum alignment of 15 amino acids.

## Data analysis

In order to reduce the sparsity of the data, low coverage samples ($n < 2,000$) were removed and annotation tables were normalized using cumulative-sum scaling (CSS) (*Paulson et al., 2013*) in Qiime 1.9.1 (*Caporaso et al., 2010*). In order to identify features (taxa or genes) that could be considered as markers of a certain treatment, data were analyzed using MetagenomeSeq (*Paulson et al., 2013*). Also, permutational multivariate analysis of variance (Adonis), non-metric multidimensional scaling and Mantel tests were performed

within the 'vegan' package in R (*Oksanen et al., 2018*). The Mantel test was performed in Qiime (*Caporaso et al., 2010*).

Network analysis was performed on the CSS normalized data using the Molecular Ecological Network Analyses Pipeline (MENAp) (*Deng et al., 2012*). Networks were constructed based on features that were present in at least 70% of the samples using the Pearson correlation matrix. Metagenomic NWs were constructed using a *p*-value cut-off of 0.01. The Gephi software (*Bastian, Heymann & Jacomy, 2009*) was used to visualize the network graphs. To determine the role of individual nodes we have applied edge degree, betweenness, Zi and Pi (*Zhou et al., 2010*) to describe the properties of each node, and plots were produced using ggplot2 (*Wickham, 2016*). Random networks were generated using the Maslov-Sneppen procedure (*Maslov & Sneppen, 2002*).

## RESULTS

### Site and leaf species effect on microbiome function and composition

The community found in the litter samples was homogeneous between plant species and sites (Fig. 1). The most common phyla were Proteobacteria (classes Gamma and Alphaproteobacteria) followed by Firmicutes (Clostridia and Bacilli) (Figs. 1A and 1B). Most of the reads detected in the libraries belonged to Bacteria. In addition to the bacterial phyla, the only phylum with normalized relative abundance above 1% was the Euryarchaeota. The functional classification of the reads was even more homogeneous than the taxonomic (Fig. 1C). The most abundant protein clusters were associated with the metabolism of carbohydrates, amino acids, and proteins.

Non-metric multidimensional scaling (NMDS) of the metagenomic data indicated that substrate (i.e., tree species) did not significantly select for a specific community (Figs. 2A and 2B). It also showed that different mangroves had a large overlap between them. This pattern was more pronounced in the NMDS based on the functional data (Fig. 2B) than the taxonomic (Fig. 2A). Two-way Adonis ($p < 0.05$) also confirmed these results. According to this test, neither site nor plant species had a significant effect on the communities' functional profile, while there is a significant effect of site on the taxonomic profile (Pseudo-F $= 2.39424$ $R2 = 0.21981$ $p = 0.020$). Despite this slightly different result, the Mantel test indicates a strong ($r = 0.86565$) and significant ($p = 0.001$) correlation between functional and taxonomic data. We also tested whether differential features in the data could be differentiated (i.e., substrate, site or substrate + site) using MetagenomeSeq (*Paulson et al., 2013*). However, no feature was significantly different between treatments.

### Network analysis

The construction of ecological networks was applied to describe the interactions between community features (i.e., taxa or genes). The interactions do not represent close contact between features but their behaviors are significantly correlated. Indeed, correlations exist between different sorts of ecological parameters (e.g., competition, mutualism, predation, environmental overlap). However, due to the complexity of microbial communities and their diminished size, the true nature of such correlations is difficult to understand.

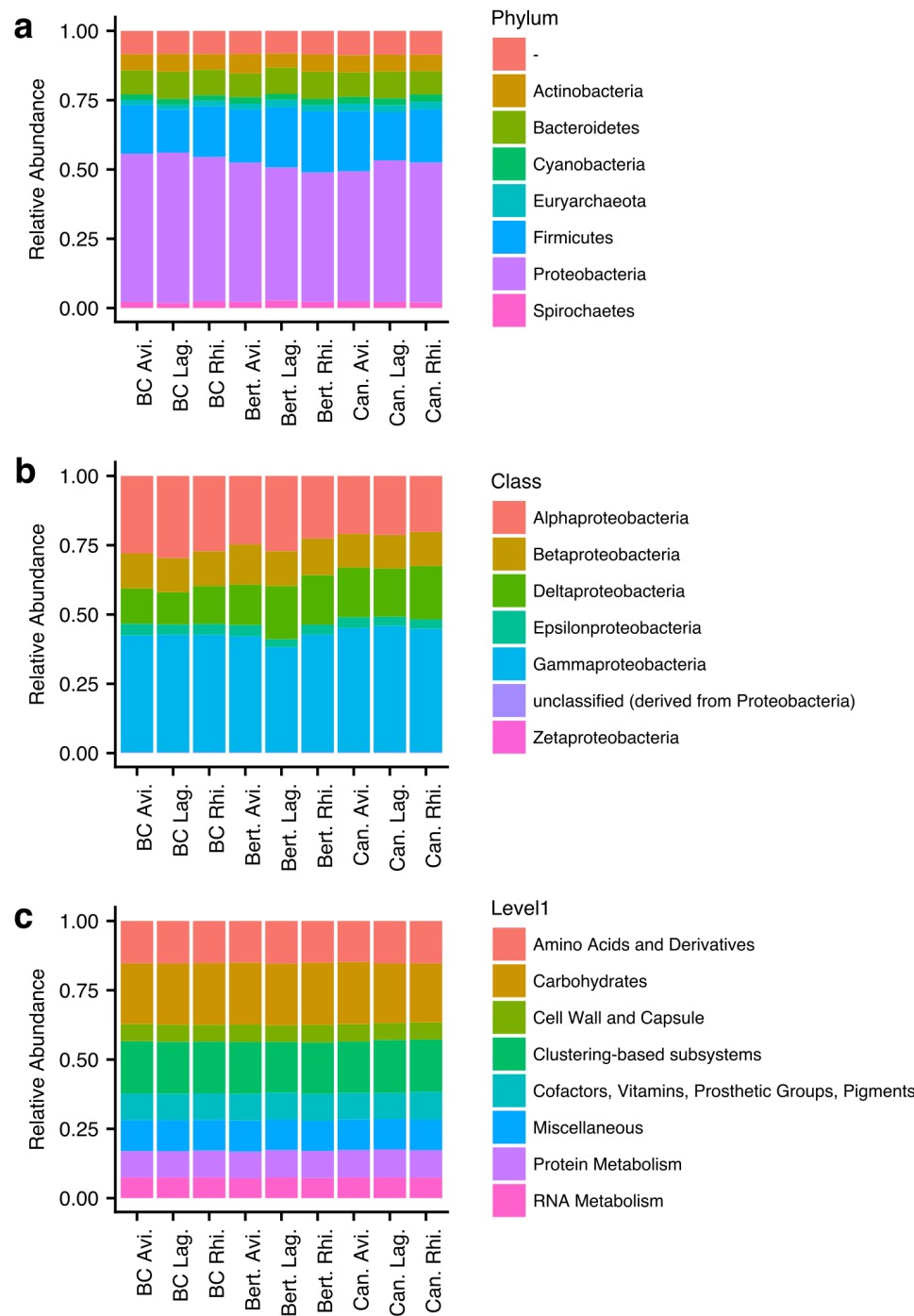

**Figure 1** **Classification of metagenomic sequences from samples of litterbags left on mangrove sediments.** (A) classification of sequences to the level of phylum; (B) classification of sequences from Proteobacteria to the level of class; (C) classification of sequences in functional SEED subsystems.

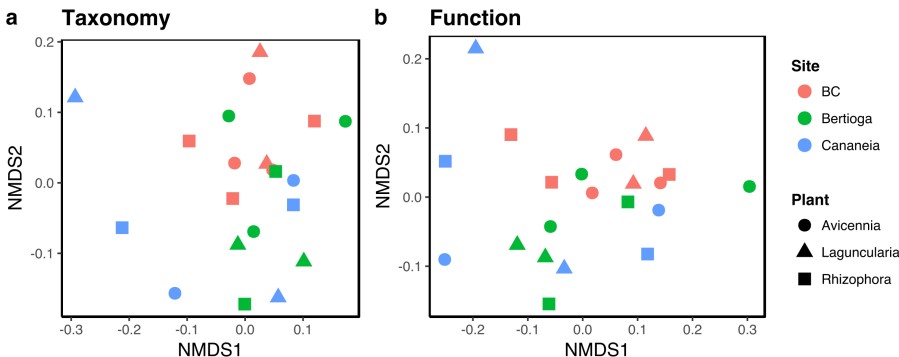

**Figure 2** **Non-metric multidimensional scaling plots (NMDS) of metagenomic data based on MG-RAST classification of sequences obtained from litterbags left on mangrove sediments.** (A) NMDS of the taxonomic classification of metagenomic data; (B) NMDS of functional classification of the metagenomic data. Samples are colored as displayed on the legend.

The network based on the taxonomic classification of the metagenomic data has revealed a complex network with 2,783 nodes and 5,754 edges (Fig. 3). This network had a high degree of modularity and several dual node subnetworks. Another feature of this NW is the high association of populations of the same phyla.

The taxonomic and functional networks exhibited scale-free characteristics, as indicated by $R^2$ of power-law fitting (0.83 for the functional network and 0.88 for the taxon network). Randomly rewiring the network connections and calculation of network properties indicated that associations observed deviate from a random association and that these networks exhibit small-world structure due to the low mean shortest path length and average clustering coefficient (Table 1).

The analysis of the centrality of individual nodes indicates that each phylum had a distinct role within this network (Fig. S2). Bacteroidetes presented the highest average Betweenness centrality (BwC) of all phyla, followed by Proteobacteria and Chloroflexi. Populations with high BwC have central positions in an NW and cannot be easily removed, whereas low BwC populations can be eliminated from the NW without disrupting the network. Another important observation is the association between taxonomic affiliation, normalized abundance, and BwC. Population with high abundance had the highest BwC and were affiliated with Bacteroidetes or Proteobacteria.

The largest subnetwork was formed by Bacteroidetes. The remaining subnetworks were divided between the other abundant phyla. Most of the subnetworks formed by Proteobacteria were separated between the different classes Proteobacteria (Fig. 4A). The ZP plot (Fig. 4B) indicates that all features present in the network are peripherals (Zi ≤ 2.5, Pi ≤ 0.62), with most of their links inside their modules. Most of them had no links outside their own modules (i.e., Pi = 0). There was only one module hub (Zi > 2.5, Pi ≤ 0.62), no connectors (Zi ≤ 2.5, Pi > 0.62) or network hubs (Zi > 2.5, Pi > 0.62). This module hub was classified as *Pseudomonas* OTU closely related to *P. aeruginosa*. This result indicates low connectivity in the network.
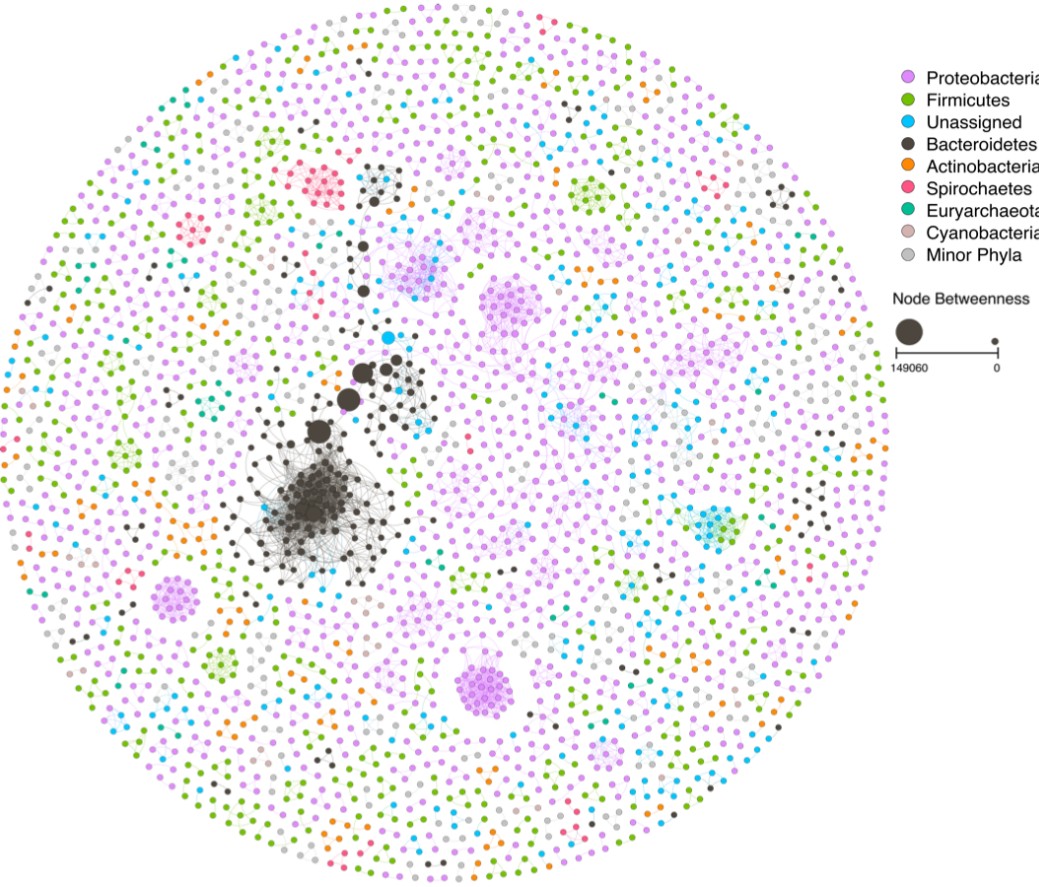

**Figure 3  Ecological network based on the taxonomic classification of the mangrove trees litter decomposition metagenomic samples.** Node size is proportional to the node betweenness. For a high-resolution version of the figure, see Fig. S1.

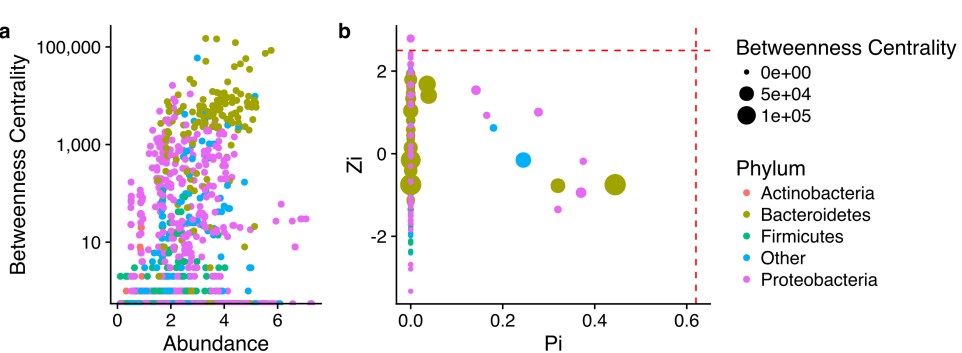

**Figure 4  Properties of each node as represented by their role within the network.** (A) Relationship between node betweeness, abundance and taxonomic assignment; (B) relationship between within-module connectivity (Zi) and among-module connectivity (Pi), node betweeness, and taxonomic assignment.

**Table 1  Indexes based on ecological network analysis of metagenomic data from decomposing leaves of mangrove trees and random trees constructed based on this data**

| Network indexes | Taxonomic | | Functional | |
|---|---|---|---|---|
| | Empirical network | 100 random networks | Empirical network | 100 random networks |
| Modularity(fast_greedy) | 0.927 | 0.490 ± 0.003 | 0.781 | 0.363 ± 0.002 |
| Lubness | 1.000 | 1.000 ± 0.000 | 1.000 | 1.000 ± 0.000 |
| Hierarchy | 0.000 | 0.000 ± 0.000 | 0.000 | 0.000 ± 0.000 |
| Efficiency | 0.827 | 0.999 ± 0.000 | 0.985 | 0.999 ± 0.000 |
| Connectedness (Con) | 0.007 | 0.851 ± 0.011 | 0.093 | 0.902 ± 0.008 |
| Transitivity (Trans) | 0.723 | 0.028 ± 0.002 | 0.453 | 0.018 ± 0.001 |
| Reciprocity | 1.000 | 1.000 ± 0.000 | 1.000 | 1.000 ± 0.000 |
| Density (D) | 0.001 | 0.001 ± 0.000 | 0.002 | 0.002 ± 0.000 |
| Centralization of eigenvector centrality (CE) | 0.171 | 0.160 ± 0.011 | 0.174 | 0.141 ± 0.011 |
| Centralization of stress centrality (CS) | 0.038 | 0.214 ± 0.014 | 18.19 | 0.220 ± 0.013 |
| Centralization of betweenness (CB) | 0.002 | 0.035 ± 0.002 | 0.009 | 0.029 ± 0.002 |
| Centralization of degree (CD) | 0.019 | 0.019 ± 0.000 | 0.021 | 0.021 ± 0.000 |
| Harmonic geodesic distance (HD) | 320.933 | 4.864 ± 0.059 | 45.375 | 4.251 ± 0.031 |
| Geodesic efficiency (E) | 0.003 | 0.206 ± 0.002 | 0.022 | 0.235 ± 0.002 |
| Average path distance (GD) | 0.03 | 3.784 ± 0.061 | 0.464 | 3.662 ± 0.036 |
| Average clustering coefficient (avgCC) | 0.524 | 0.012 ± 0.002 | 0.636 | 0.012 ± 0.001 |

The ecological network constructed from the functional assignment of the metagenomic sequences show a larger and more complex net of interconnected nodes (Fig. 5) with 4,030 nodes and 12,648 edges. The clustering by classification is not apparent in this network. Randomly rewiring the network connections and calculation of network properties indicate that associations observed deviate from a random association (Table 1).

The functional network was highly connected, indicating a strong redundancy of nodes. As such, it was difficult to identify among the most frequent functional groups one with highest BwC (Fig. S4).

The functional network shows a clear relationship between abundance and BwC, however this is not the case at a taxonomy level (Fig. 6A). However, nodes with higher BwC had a central role in the network (as module hub, connectors or network hubs) (Fig. 6B). The ZP plot (Fig. 6B) indicates that most of the features present in the network are peripherals ($Zi \leq 2.5$, $Pi \leq 0.62$), with most of their links inside their modules. However, links with high BwC held important positions in these networks as module hubs ($Zi > 2.5$, $Pi \leq 0.62$) and connectors ($Zi \leq 2.5$, $Pi > 0.62$). Additionally, no network hub ($Zi > 2.5$, $Pi > 0.62$) was present in these networks. This result ndicates a low connectivity in the network with a lot of small independent modules.

## DISCUSSION

Environmental dynamics pose a challenge to the survival of nutrient cycling organisms in estuarine environments (*Holguin, Vazquez & Bashan, 2001*). In the case of mangroves, sediments can be dry or submerged as well as subjected to fresh or marine environments
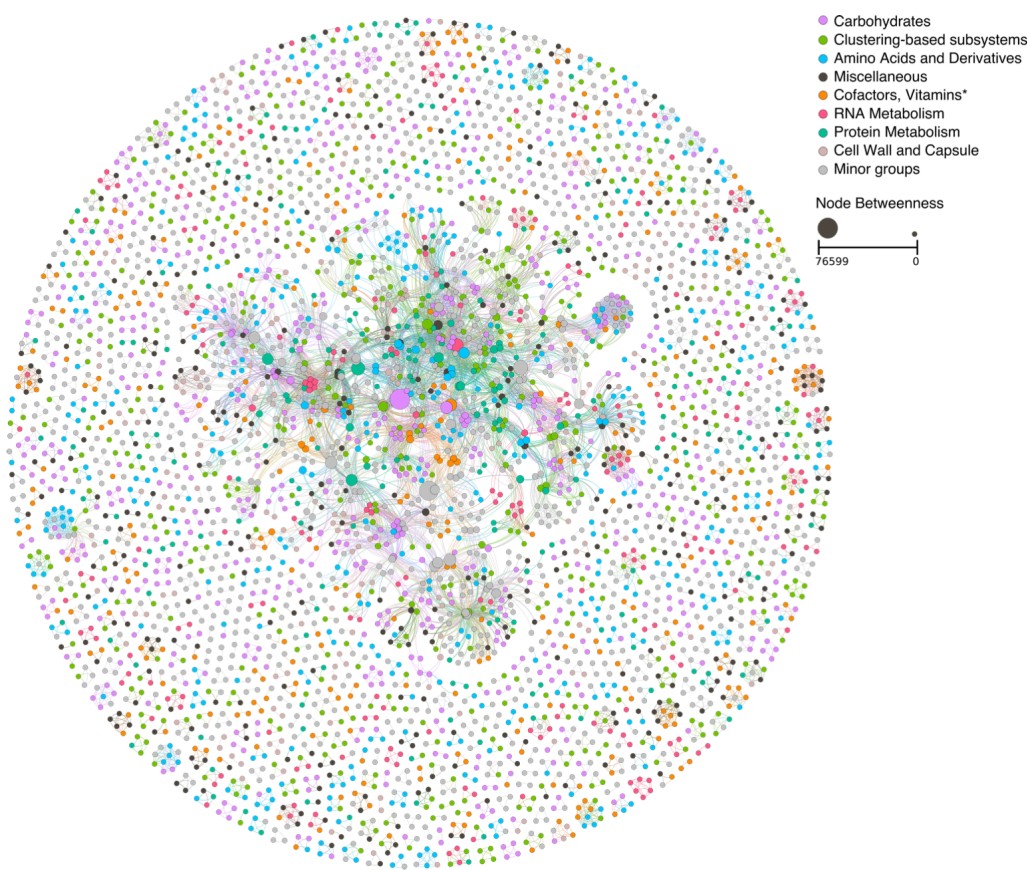

**Figure 5** **Ecological network based on the functional classification of the mangrove trees litter decomposition metagenomic samples.** Node size is proportional to the edge betweenness. For a high-resolution version of the figure, see Fig. S3.

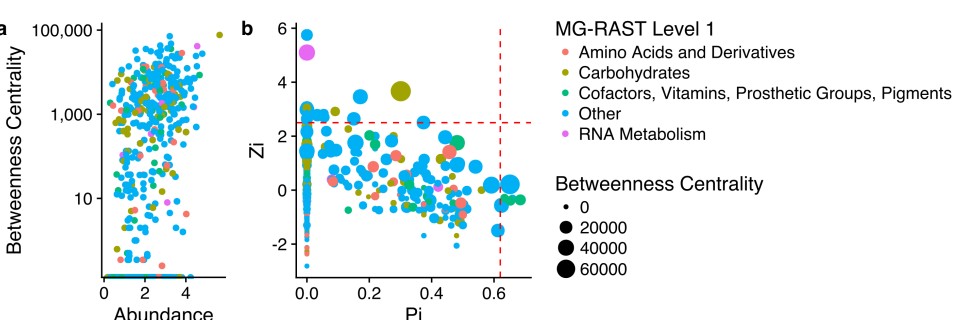

**Figure 6** **Properties of each node as represented by their role within the network.** (A) Relationship between node betweeness, abundance and functional assignment; (B) relationship between within-module connectivity (Zi) and among-module connectivity (Pi), node betweenness, and functional assignment.

(*Bouillon et al., 2004*), which results in a complex microbial assemblage (*Freschet et al., 2013*; *Miura et al., 2015*; *Moitinho et al., 2018*). In this study, we have applied litterbag experiments to unravel the effects that plant species has on the microorganisms that colonize their decaying leaves and to identify how the environmental characteristics affect this process. Interestingly, the community composition did not present high variation when we looked at broader taxonomic ranks (such as phylum and class). This apparent stability was observed regardless of the factor analyzed (i.e., plant species or mangrove site). This effect was stronger in the functional than taxonomic classification. However, the contrasting pattern between functional and taxonomic classification is relatively common and has been observed in many environments (*Costello et al., 2012*; *Delmont et al., 2012*; *Taketani et al., 2014*). Furthermore, the communities found in the decaying leaves were different from those usually found in mangrove sediments that have a high abundance of sulfur reducing Deltaproteobacteria (*Andreote et al., 2012*; *Varon-Lopez et al., 2014*) while leaves were dominated by Gamma and Alphaproteobacteria. This must be determined by the fact that the environment in which the decomposition takes place is not suitable for these organisms due to the higher concentration of $O_2$ which also prevents the presence of methanogenic archaeal populations (*Dias et al., 2011*; *Mendes et al., 2012*). This suggests that these organisms may come from aerobic sources such as air, water, and leaf.

The small variation in the composition reflected in NMDS and Adonis patterns which were found to be not significant. This is indicative that despite the variation in environmental characteristics that the community profile is quite stable. Alternatively, we can propose that the populations that inhabit this material might be selected to withstand this variation.

Fluctuations of fresh and marine waters in estuarine ecosystems result in spatial and temporal variation of microbial communities (*Guo et al., 2017*), as such, mangrove litter is subjected to a large range of biotic and abiotic environmental factors.

Plant material with different chemical properties has been shown to have only a minor effect on the bacterial community composition (*Tláskal et al., 2018*). This explanation is supported in our study as we did not find any functional feature or taxonomic group that was differentially abundant in any leaf species or mangrove site.

The functional and taxonomic networks presented a great number of co-occurring nodes. The network constructed based on these data presented scale-free characteristics and this type of network is considered very resistant to disturbances and the removal of nodes (*Green et al., 2017*), and indicates a relatively stable microbial community structure.

These networks also exhibit small-world structure, which indicates that nodes are accessible to every other node through a short path (*Layeghifard, Hwang & Guttman, 2017*). These networks are believed to be highly coordinated while allowing for a high degree of functional specialization into clustered units (*Watts & Strogatz, 1998*; *Green et al., 2017*). However, a small-world structure is common in large networks (*Green et al., 2017*).

The taxonomy based network formation indicates that there is a tight link between phylogeny and lifestyle since the co-occurrence patterns indicate a preference for similar environmental conditions (Fig. 4A). The correlations between nodes of the same taxonomic groups might be related to similar lifestyles shared by closely related taxa

(*Philippot et al., 2010*). Despite the possibility that minor differences between such taxa might lead to distinct ecological strategies or lifestyles (*Fraser et al., 2009*; *Denef et al., 2010*), it can be speculated that there is some degree of redundancy in this networks which would aid in the stability of the process.

The Bacteroidetes are recognized as consumers of complex polysaccharides in marine environments and their genomes have a large number of genes related to glycoside hydrolase (GH) families (*Bauer et al., 2006*). Hence, these bacteria might have an important role in the leaf degradation despite the expected role of fungi in this process (*Hu et al., 2017*; *Tláskal et al., 2018*). This result indicates that in mangrove sediments, bacteria (especially Bacteroidetes) might have an important role in the decomposition, possibly due to the lower cost of reproduction of this bacterial taxa, that are considered r-strategists (*Hu et al., 2017*), in the energy limited anaerobic sediments (*Taketani et al., 2010b*).

The second phylum with the highest BwC were the Proteobacteria which is a very versatile group (*Cobo-Simón & Tamames, 2017*) and very abundant in marine environments and mangroves (*Taketani et al., 2010a*; *Andreote et al., 2012*; *Varon-Lopez et al., 2014*). In terrestrial ecosystems, Alpha-, Beta- and Gammaproteobacteria were found to be prevalent in the initial phases of litter degradation due to their fast growth (*DeAngelis et al., 2013*) and they also become more prominent over time in phyllospheric communities (*Vojtěch, Vorískivá & Baldrian, 2016*). This wide range of lifestyles contributed to the broad dispersal of BwC observed in Fig. 4.

The topological role of individual nodes (Zi-Pi plot) indicated that a *Pseudomonas* (Gammaproteobacteria) is the only taxon that has an important position in this network as a module hub. Hubs have a central role in a network and/or module (*Jiang et al., 2015*). Hence, this pseudomonad is a key node within a module despite its low abundance and BwC. However, scale-free networks usually display only a small portion of hubs (*Green et al., 2017*) which contributes to its robustness.

The role of individual nodes in the functional network was slightly different than observed in the taxonomic analysis. All of the broad functional groups had a similar average BwC which indicates that they have similar importance within the network. Besides, nodes with higher BwC were identified as hubs (module hubs and connectors) which indicates that the removal of these would affect the structure of the network (*Deng et al., 2012*). However, since within these nodes there is a mixture of different taxa that are likely to respond differently to perturbation, there is a chance that the higher robustness of the taxonomic network would aid the community to endure stresses. Hence, there might be an important role of functional redundancy to the stability of the community present in mangrove litter (*Strickland et al., 2009*; *Banerjee et al., 2016*) which would aid in maintaining the efficient decomposition of litter (*Kaiser et al., 2014*)

## CONCLUSIONS

This study has shown that the community present in mangrove plant's decaying material is stable despite differences in plant species or mangrove location. These communities form a tight network that is robust and resistant to disturbances and therefore capable of withstanding the constantly changing environment that mangrove ecosystems present.

## ACKNOWLEDGEMENTS

The authors thank João Luiz da Silva, Vanessa Nessner Kavamura, Natália Franco Taketani and Fabio Sérgio Paulino Silva for their support during sampling.

### Funding

This work was supported by FAPESP (2013/03158-4). Rodrigo G. Taketani received a FAPESP Young Investigator Fellowship (2013/23470-2). The funders had no role in study design, data collection and analysis, decision to publish, or preparation of the manuscript.

### Grant Disclosures

The following grant information was disclosed by the authors:
FAPESP: 2013/03158-4.
FAPESP Young Investigator Fellowship: 2013/23470-2.

### Competing Interests

The authors declare there are no competing interests.

### Author Contributions

- Rodrigo G. Taketani conceived and designed the experiments, performed the experiments, analyzed the data, contributed reagents/materials/analysis tools, prepared figures and/or tables, authored or reviewed drafts of the paper, approved the final draft.
- Marta A. Moitinho performed the experiments, approved the final draft.
- Tim H. Mauchline authored or reviewed drafts of the paper, approved the final draft.
- Itamar S. Melo conceived and designed the experiments, contributed reagents/materials/analysis tools, authored or reviewed drafts of the paper, approved the final draft.

### Field Study Permissions

The following information was supplied relating to field study approvals (i.e., approving body and any reference numbers):

Field sampling was approved by the System of authorization and information in biodiversity (20366-3).

### Data Availability

Metagenomic sequences are available in MG-RAST, project number (mgp13300).

### Supplemental Information

Supplemental information for this article can be found online at http://dx.doi.org/10.7717/peerj.5710#supplemental-information.

![PeerJ]

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
