# Peer review of "Co-occurrence patterns of litter decomposing communities in mangroves indicate a robust community resistant to disturbances"

_PeerJ, doi:10.7717/peerj.5710_

## Round 0.1 · original submission · Major Revisions

Dear authors,

You will find that the reviewers have highlighted certain areas where a revision will help clarify a number of outstanding issues. In general, the reviewers agree that your piece of work is valuable and of interest to the community. In particular, I highlight the need to be more precise in your description of the experimental conditions and of the methodology employed for the analyses. Please also consider carefully all the commentaries about lack of clarity in your introduction and discussion sections and the need for a careful revision of the English language.

Reviewer 1 ·

Basic reporting

no comment

Experimental design

no comment

Validity of the findings

no comment

Additional comments

The article entitled: “Co-occurence patterns indicate a robust community resistant disturbances during litter decomposition in mangroves” was well done and showed interesting results. However, some modifications are necessary to improve the quality of the paper.

General view:

- Very important: it is important to show the hypotheses!
- English need to be improve
- The text is fragmented on many parts (highlighted for section on next commentaries). There are paragraphs so big and others so short. This situation compromises the reading of the text. Please, improve the writing.
- There are many errors of typing and spelling (highlighted for section). Please, correct them.

Abstract:

- This section is ok. The description of the ideas and the methodologies was well done. However, the discussion needs to be improved. It is important to add a strong take home message! This part is weak.

Introduction:

- Line 40-46: Text fragmented. It is important to become the text more fluid. It could be interesting add information about primary productivity on mangrove sediments on introduction. The litter consumption is the mainly processes associated with carbon cycle on this biome. But this processes in not unique. Please, make mentions about primary productivity on mangroves.

- Line 47-49: This paragraph is exactly the same on abstract. Please, modify.

Material and Methods:

- All collected leaves showed similar phenological stages? Add this information.

- Line 83: Please, add more information about the similarity and/or differences observed on plant species. It is important for understand the possible differences found on organic matter source for microbial decomposition.

- Line 86: The experiment was conducted on replicate? How many replicates? Please add this information

- Line 93: “DNA quality and quantity was evaluated in Nanodrop 2000.” Is the integrity checked on Agarose gel?

- Line 98: As the interest of the authors was to study litter decomposition, why the authors didn’t use others database as CAZY (carbohydrate active enzyme) to upgrade its analysis?

Results:
- Line 129: Please check and correct this phrase – “the only Archaea phylum with normalized relative abundance above 1% was the Euryarchaeota”.

- Line 132: Mixed groups (Clustering based subsystems and Miscellaneous) is a designation that Subsystem curators used to denote experimental system that are works in progress. They should not be included in a manuscript for publication as there are not verified. Should be removed from the analysis.

- Line 151: Hoiwever shift for However

- Line 157: This phrase is displaced. Both? Which? This paragraph could be add on the beginning of line 153. Both (taxonomic/functional) ….

- Line 164: Change proteobacteria and clroloflexi to Proteobacteria and Chloroflexi

- Line 172: proteobacteria to Proteobacteria

- Line 179: Please, put the word Pseudomonas in italic

Discussion:

- Line 204: There are another publications on this thematic. The authors could add these references together with unpublished data

- Lie 229: This sentence justify the add of information about plant characteristics

Reviewer 2 ·

Basic reporting

I believe that the use of scientific English could be slightly improved. There are a few lines with too colloquial statements. But in general, the article can be easily followed.

Sufficient background is generally provided. The article follows the standard structure and raw data is shared, but a number of notes are needed:

The introduction is generally well written. I'd add a short note about how co-occurrence patterns may be important in the mangrove ecosystem, in addition to the already shown general importance. Then, this should be clearly linked to the conclusions section, which now is somewhat disconnected.

In the results, there are a number of lines were the authors tend to partially discuss their results. For instance: L147-L152. This should be corrected and avoided here and through.

I think the way results are presented and the figures need to be greatly improved.

Experimental design

Sampling strategy is adequate. In the methods section, other than specifying the strategic factors (i.e., different mangrooves), it should be stated how many samples were sequenced. Right now it is unclear, and it is crucial to understand the rest of the analysis.

Studying co-occurrence patterns through network analysis is timely and the research question has interest. However, I have missed on how the results of the article fit into the existant litterature regarding litter decomposition, rather than a more general framework (which is ok, but not enough).

Validity of the findings

Again, I think the conclusion could be better connected to the introduction. I think that some speculation and interpretation could be better phrased to be better identified as speculation.

Additional comments

Introduction
L18: change to "macro- and microorganisms"
L26-27: use capital letters in bacterial taxa
L31: low MSPL and high clustering coefficient. Otherwise, it states that both should be low.
L57: different -> specific
L73: scale -> scales

(Methods)
L97. I'd add here the mean length of retrieved sequences. Otherwise, it could be added in the next section.
L113-114. Odd sentence. Rephrase.
L120. About node properties. I recommend describing these more to better understand their ecological significance.

(Results)
L126. Here and through, bacterial taxa should be in capital letters.
L151. Hoiwever -> However.
L151. Also, add environmental overlap as an option between brackets.
L155. Striking. Why striking? These evaluations should be saved for discussion and supported accordingly. The fact that some taxa within the same phyla are associated could be explained by functional redundancy or environmental overlap, through a phylogenetic conservation of traits. Develop on this within the discussion.
L173-175. Again, discussion.

(Discussion)
L208: 'so much'. Avoid colloquialism here and through.
L214. Rephrase.
L220. Rephrase.
L232. 'lead us to propose'. I believe that the level of co-occurrence is a fact based on the shown results. I'd expand on why is this pattern found.
L253-L256. Discussing ecology of Proteobacteria in 4 lines lacks ecological significance. Again, split at least classes within Proteobacteria.
L275. i.e. plant ... -> remove i.e and state factors used in the study. Also, I believe that 'differences in the environment' is not accurate since there is not a report of environmental variables. Maybe habitat or site could be more appropriate.


(Figures)
FigS1, S3, 3: and 6 I do not think it is required to duplicate a figure in the supplementary documents.
FigS2. I'd replace the scientific notation in the axis.Also, the first boxplot has no legend. In addition, to make the figure clearer I'd shrink dot size. Also, I think it would be of interest to split Proteobacteria into its main classes. Also, to improve clarity and significance of the figure, I'd optionally remove those bacterial groups were Betweenness is 0, specifying so in the legend.
Fig S2 and S4: maybe reorder the x-axis by mean/median to improve clarity.

Fig1: are these barplots per sample? If so, I understand that the number of samples taken was 21?. Since there are few differences per sample, i'd consider to regroup these into factor levels through averaging profiles (i.e., one single barplot for Avicenia, other for Laguncularia...). In the B panel, I recommend to split Proteobacteria into its main classes (i.e., alpha, beta, gamma...).
Fig2. Are these nMDS done solely with the annotated orfs?, I think further clarity should be provided.
Fig 4. Definitely, legend should be improved. I think that you could group many taxa into an 'others' category to improve the visibility of the figure. For example, all those taxa with < 2 nodes, but I'd leave this to the author's criteria.
Fig 5. Same philosophy than Fig 4. Here I see two options. Or group some categories into an 'others' cluster (therefore, reducing the number of colors to 4-8), or simply removing colors if they want to highlight the abundance-betweenness relationship.

Reviewer 3 ·

Basic reporting

- In this manuscript the authors investigated the bacterial community associated with mangrove leaves decomposition in mangrove sediments in Brazil. Although the main issue is very interesting, with the possibility to generate new information regarding the role of microbial communities in the decomposition, the dataset is poorly explored and no novelty is presented.
- My main concern is related to the lack of hypothesis. What is the main hypothesis of this study? The manuscript is merely descriptive and focused only on the results of the network (please see comments below).
- The authors did not find differences between the treatments for both taxonomy and functions, and I have some concerns about these results. First, it’s not clear what is the taxonomic and functional level used for the analysis. Reading the paper it seems they analyzed at phylum/class level. In order to get differences I would suggest to go deeper in the analysis.
- Why the analysis was conducted only after 60 days? In order to assess the community dynamics would be adequate to at least have the point zero analysis, to compare the beginning to the end of the chronosequence. Maybe no difference was found because the community was already established after the 60 days. Also, in order to detect the selection of specific groups in the decomposition of the plant material, a comparison with the community of the mangrove sediment is necessary.
- Considering that the authors are trying to assess the community related to decomposition, specific microbial groups and functions should be the focus of the analysis. With the metagenome data the authors could focus on functions related to decomposition and show what is happening comparing the different treatments.
- The most part of the manuscript is regarding the network, however there are several concerns about how the authors conducted this analysis. Why there is only one network? Considering three different mangroves and three different plants I would expect several networks to compare the community dynamics between the treatments.
- The figures are wrongly numbered and make the reading a little bit confused.
- Small typing errors can be found throughout the manuscript. A careful revision is needed.
- What is the possible effect of the bags used in the experiments? Please discuss this issue.
- Please use different symbols in NMDS figure for a better visualization of the patterns.

Experimental design

The experimental design description is very shallow, with the absence of important information to support the analysis and the results. For example:
- What is the location of these mangrove sites? Please provide geographical location.
- Please provide more details about the sampling: how many bags were used per mangrove site? How many samples of DNA were extracted? How many samples were sequenced?
- How could the bag affect the decomposition?
- How many samples were used for network reconstruction?
- Why the samples were grouped for network reconstruction?
- In which taxonomic and functional level were analyzed the data?
- Were the three different plant material mixed?
- Why only one sampling time (60 days)?

Validity of the findings

Considering all the problems raised in the comments above, the findings are not supported by the experimental design and methodology applied. However, considering the dataset I encourage the authors to re-run the analysis and re-organize the manuscript.

---

## Round 0.2 · Minor Revisions

Many thanks for implementing many of the changes requested by the referees. The effort made has certainly improved your article.
Please note however that Avicennia is mispelled in legend of Figure 2 (Avicenia). Please correct the text and figure.

Reviewer 1 ·

Basic reporting

no comment

Experimental design

no comment

Validity of the findings

no comment

Additional comments

The authors have addressed adroitly the points raised in the review.

Reviewer 2 ·

Basic reporting

Figure use has greatly improved.

Experimental design

Sampling is now better explained.

Validity of the findings

The authors may be wary of linking cause to consequence in their dissertation. However, the experimental setting is strong and controlled, so findings are still relevant.

Additional comments

The manuscript has improved. However, authors should go carefully through the text once more, maybe with the help of an english speaker (or online tools such as grammarly) to make sure grammar is correct. It is understandable that language style varies, but grammar should be correct since English is the standard for science.

---

## Round 0.3 · Minor Revisions

Dear authors,

After a second round of revisions, the referees agreed that your contribution has significantly improved. One of the reviewers remains of the opinion that the quality of your English writing needs to be improved. I am therefore recommending minor revisions and expect that you will be approaching a native English speaker to improve the quality of your presentation.

---

## Round 0.4 · accepted · Accept

The authors have added an author that has helped with the language. There are still a few spelling and grammatical errors in the text, so it would be good if the authors looked carefully through the document one more time.

#